# Insights into centriole geometry revealed by cryotomography of doublet and triplet centrioles

**Garrett A Greenan[1,2,3], Bettina Keszthelyi[1,3], Ronald D Vale[2,3], David A Agard[1,3]***

[1]Department of Biochemistry and Biophysics, University of California, San Francisco, United States; [2]Department of Cellular and Molecular Pharmacology, University of California, San Francisco, United States; [3]Howard Hughes Medical Institute, San Francisco, United States

**Abstract** Centrioles are cylindrical assemblies comprised of 9 singlet, doublet, or triplet microtubules, essential for the formation of motile and sensory cilia. While the structure of the cilium is being defined at increasing resolution, centriolar structure remains poorly understood. Here, we used electron cryo-tomography to determine the structure of mammalian (triplet) and *Drosophila* (doublet) centrioles. Mammalian centrioles have two distinct domains: a 200 nm proximal core region connected by A-C linkers, and a distal domain where the C-tubule is incomplete and a pair of novel linkages stabilize the assembly producing a geometry more closely resembling the ciliary axoneme. *Drosophila* centrioles resemble the mammalian core, but with their doublet microtubules linked through the A tubules. The commonality of core-region length, and the abrupt transition in mammalian centrioles, suggests a conserved length-setting mechanism. The unexpected linker diversity suggests how unique centriolar architectures arise in different tissues and organisms.
DOI: https://doi.org/10.7554/eLife.36851.001

*For correspondence:
agard@msg.ucsf.edu

**Competing interests:** The authors declare that no competing interests exist.

## Introduction

The microtubule-based centriole is a complex organelle that plays fundamental roles in cell division, motility and signaling. When cell division is favored, centrioles accumulate thousands of copies of hundreds of different proteins to amass the pericentriolar material (PCM). From this PCM, microtubules are nucleated to form the mitotic spindle that facilitates accurate chromosome segregation (*Wittmann et al., 2001*). Under conditions where division is not favored, or the cell is terminally differentiated, a ciliogenesis program is initiated (*Ishikawa and Marshall, 2011*). During ciliogenesis, the centriole moves from the cell center to the plasma membrane to become a basal body. Doublet microtubules elongate from the distal end of the basal body to form the axoneme, which extends beyond the cell periphery covered in plasma membrane to form a specialized compartment called the cilium. These nine, doublet microtubules provide a structural scaffold, and are essential for transport within the ciliary compartment, which can be up to several micrometers in length (*Stepanek and Pigino, 2016*; *Ishikawa and Marshall, 2011*). Depending on the needs of the cell, the cilium can be either a motile cilium, like those that propel sperm or move fluid in the airway, or an immotile sensory cilium, like the light-sensitive compartment of photoreceptor cells or cilia that participate in tissue patterning via the Hedgehog signal-transduction pathway. Genetic mutations that result in structurally defective axonemes, such as those lacking axonemal dyneins, result in defective cilia and give rise to a group of human diseases called ciliopathies (*Reiter and Leroux, 2017*). The wide range of phenotypes associated with ciliopathies reveal the important role that axoneme structure and cilium integrity play in human development.

In all species in which centrioles have been defined, centrioles are comprised of microtubules arranged in a 9-fold symmetrical cylinder. During centriole duplication in S phase, a 9-fold symmetric structure called the cartwheel is assembled that defines the fundamental centriole geometry (*Azimzadeh and Marshall, 2010*). The cartwheel is composed of nine symmetrically arranged SAS-6 homodimers that attach to the microtubules of the centriole through a structure called the pinhead (*van Breugel et al., 2011*; *Kitagawa et al., 2011*). At the end of mitosis, many cartwheel proteins are no longer present at the centriole, presumably because the cartwheel has been degraded (*Vorobjev and Chentsov, 1980*; *Arquint and Nigg, 2014*). While all organisms that have centrioles use SAS-6 to set the 9-fold geometry, the microtubules that comprise the centriole cylinder can be singlet, doublet or triplet microtubules depending on species, and the structures that the centriole will nucleate in the cell. In the early *Caenorhabditis elegans* embryo, centrioles are comprised of 9 singlet-microtubule (SMT) rods surrounding a central cylinder and non-motile cilia are formed in sensory neurons after the singlet centriole has been converted to a doublet basal body (*Chalfie and Thomson, 1979*; *Pelletier et al., 2006*). Most *Drosophila* cells have full or incomplete doublet-microtubule (DMT) centrioles, although singlets seem to dominate in the early embryo, which has very rapid cell cycles (*Bate, 2009*). DMT centrioles template the formation of nonmotile cilia in the sensory organ precursor (*Hartenstein and Posakony, 1989*). However, in the male germline stem cells, the doublet centriole is converted to a triplet basal body, and it is this triplet basal body that facilitates the formation of a motile axoneme (*Gottardo et al., 2015*). In mammals, the triplet centriole is the default state, and depending on cell fate, triplet centrioles facilitate the formation of the appropriate cilium type.

The relationships between the microtubules of the centriole and the axoneme are poorly understood. Data from plastic-embedded sections have shown that centrioles in mammalian cells are composed of angled TMTs that give rise to axonemes with inline DMTs (*Jana et al., 2016*; *Paintrand et al., 1992*). However, it is unclear how this change in geometry is achieved. Electron cryo-tomography studies of *Chlamydomonas* basal bodies have shown that the microtubules of the axoneme are contiguous with those of the basal body and have also revealed numerous non-tubulin decorations (*Li et al., 2012*). However, the lack of high-resolution structural data of centrioles prior to ciliogenesis has hampered our understanding of centriole biogenesis and axonemal templating. Recent technological advances have pushed the attainable resolution of single-particle electron cryo-microscopy (cryoEM) to rival that of X-ray crystallography (*Paulsen et al., 2015*). These single-particle cryoEM methods have been applied to solve the structure of isolated axonemes at sub-nanometer resolution, revealing new aspects of its structure and decoration (*Ichikawa et al., 2017*). Such high resolution is facilitated by the structural rigidity and extreme length of axonemes, greatly enhancing the number of units that can be averaged. However, the contextual information and information on the connection between the axonemes and the basal bodies is lost upon isolation. By contrast, electron cryo-tomography (cryoET) combined with sub-volume averaging can maintain all relevant spatial information, albeit at the expense of resolution due to a thicker sample and typically reduced potential for averaging (1000 s of copies of the relevant structural unit vs. 100,000 s) (*Mahamid et al., 2016*; *Kourkoutis et al., 2012*).

Here, we focus on defining the structure of centrioles purified from Chinese Hamster Ovary (CHO) or *Drosophila melanogaster* S2 cells by cryoET. Our data show that, although mammalian and fly centrioles have a broadly similar quasi-9-fold symmetrical cylindrical architecture, the linkages between centriole microtubules that achieve this architecture are completely different. Furthermore, the longer mammalian centriole is organized into two distinct domains along its proximal-distal axis: a proximal domain, whose composition of interconnected triplets shares geometry with the fly centriole, and a distal domain comprised of novel linkages between centriole microtubules, generating a geometry that is intermediate between that of the proximal centriole and an axoneme. Together our data suggest a model where the distal mammalian centriole provides a transitional region that allows for the templating of both motile and sensory axonemes, a region and potential that does not exist in somatic fly centrioles.

## Results

### Mammalian centriole structure

We first examined the overall structure of mammalian centrioles in ice. Purified CHO centrosomes were imaged using electron cryo-tomography (cryoET), and tomograms of centrioles were generated. These CHO centrioles were on average 440 nm long, and appeared to taper along their length, being wider at the proximal end, which would be connected to the central hub, and narrower at the distal end which, in a basal body, would face the membrane (*Figure 1a*). Our tomograms were feature rich, and showed centriolar microtubules covered in electron dense pericentriolar material (PCM) from which centrosomal microtubules are nucleated (*Figure 1a*).

As expected, the raw data indicated that CHO centrioles were composed of interconnected TMTs. We did not observe any distal or subdistal appendages decorating the TMTs, and were thus unable to use appendages to distinguish mother from daughter centrioles in our tomograms. To determine the structure of the TMTs, disc-shaped volumes 3 tubulin heterodimers in height (*Figure 1—figure supplement 1*) were extracted along the length of each individual TMT rod, for each centriole (N = 9) (see Materials and methods). The choice of a 3 tubulin (24 nm) segment length is somewhat arbitrary, but was chosen as a tradeoff between maximizing the number of segments (better averages) but still reliably allowing for the discovery of structures having 4, 8, and 16 nm repeats.

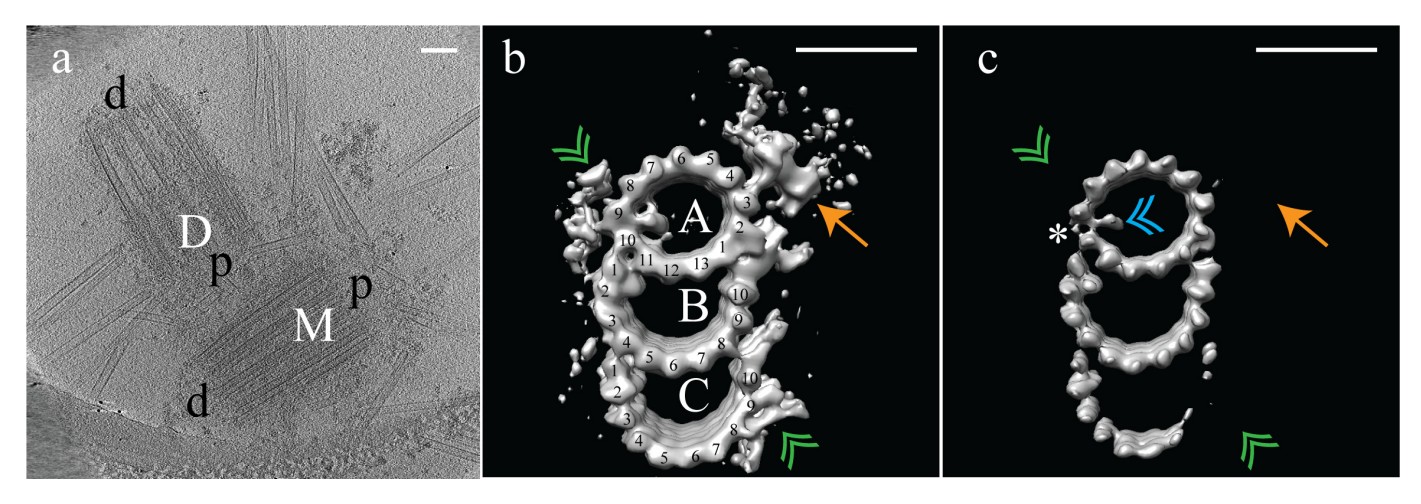

**Figure 1.** Electron cryo-tomography of mammalian centrioles. (a) Chinese hamster ovary (CHO) centrioles were imaged using electron cryo-tomography. The mother centriole [M] was decorated with peri-centriolar material along its entire length, while the daughter centriole [D] appeared less well decorated, especially at the distal end. Proximal and distal ends of the centriole are denoted 'p' and 'd', respectively. (b) Sub-volume averaging of CHO centrioles generated a map with triplet microtubules, consisting of a complete 13-protofilament A-tubule, and 10-protofilament B- and C-tubules. Both parts of the A-C linker (green double arrowheads) that connects adjacent triplets were obvious in the map, as was the pinhead (gold arrow) attached to protofilament A03. Densities binding to the lumenal surface of the microtubules were seen binding protofilaments A09/A10, B01/B02, and C01/C02. (c) High contouring of the triplet map showed the core structures that were present in the majority of all sub volumes. The absence of the A-C linker and pinhead indicated that they were at less than 100% occupancy in the map. The asterisk (*) denotes the likely site of the microtubule seam between protofilaments A09 and A10. This gap between protofilaments A09 and A10 was straddled by an A-tubule MIP (blue double arrowhead), a MIP that was present at high occupancy in the map. Scale bars are 100 nm in (a), and 25 nm in (b) and (c).

DOI: https://doi.org/10.7554/eLife.36851.002

The following figure supplements are available for figure 1:

**Figure supplement 1.** Checking for centriole periodicities.
DOI: https://doi.org/10.7554/eLife.36851.003

**Figure supplement 2.** Gold standard FSC resolution determination.
DOI: https://doi.org/10.7554/eLife.36851.004

**Figure supplement 3.** ResMap calculation of resolution for the whole-population CHO average.
DOI: https://doi.org/10.7554/eLife.36851.005

**Figure supplement 4.** CHO centrioles were flattened within the ice sheet.
DOI: https://doi.org/10.7554/eLife.36851.006

This produced a dataset that contained ~2100 unique subvolumes, which were aligned to a common reference (see Methods) and averaged to generate a global map of the CHO TMT (*Figure 1b*) (EMD-7775) at ~3.5 nm resolution (*Figure 1—figure supplement 2*) although local resolution metrics indicate that the majority of the volume has a resolution between 2.4 and 2.8 nm (*Figure 1—figure supplement 3*). Consistent with previously published data from basal bodies (*Guichard et al., 2013*; *Li et al., 2012*), the CHO centriole consisted of a TMT having a 13 protofilament A-tubule and 10 protofilament B- and C-tubules. The external surface of the global map showed two large densities - the 'pinhead' (*Figure 1b* - orange arrow) that bound to protofilament A03, and both parts of theA-C linker' (*Figure 1b* - green double arrowheads) a structure known to bridge adjacent TMTs through the C08/C09 and A09 protofilaments (*Winey and O'Toole, 2014*). Additionally, external densities were evident in three clusters near A01/A02-B10 (the inner A-B junction), A09-A10 and B08-C10 (the inner BC-junction). Given the high degree of structural preservation using cryoET, the lack of cart-wheel spokes attaching to the pinhead is in agreement with published work suggesting that the cart-wheel is specific to procentrioles and removed during the progression from mitosis to interphase (*Vorobjev and Chentsov, 1980*; *Arquint and Nigg, 2014*).

At this resolution, we observed four microtubule-internal proteins (MIPs) on the luminal surfaces of the TMTs (*Figure 1b*). The most prominent was within the A-tubule, straddling a gap between the A09 and A10 protofilaments (*Figure 1b and c*), a gap that was much larger than between any of the other protofilaments of the A-tubule (asterisk in *Figure 1c*). Recent single-particle cryoEM data on cilia assigned the microtubule seam between the A09 and A10 protofilaments (*Ichikawa et al., 2017*). This suggested that the large A-tubule MIP is binding along the internal surface of the A-tubule at the unique microtubule seam. This roughly two-fold symmetric MIP connects diagonally across the seam, although whether the connection is to the two adjacent α- or β-tubulin subunits cannot be determined at this resolution. A second, smaller A-tubule MIP appeared to bind directly to protofilament A11. MIPs also bound to the luminal surfaces of the B- and C-tubules, bridging the B01-B02 and C01-C02 protofilaments, respectively (*Figure 1b*). While the function of the B- and C-tubule MIPs is unknown, their similarity in shape and positioning suggest a common mechanism likely important for defining or stabilizing the outer AB- and BC-junctions.

To regenerate the 9-fold symmetry of the centriole, the average was fit back into the raw data using the refined coordinates and angles from the sub-volume alignment. This refitting revealed that all centrioles in our dataset were significantly flattened. While the degree of flattening varied between datasets, some centrioles more closely resembled two parallel arrays of triplet microtubules than the stereotypical 9-fold symmetric structure that was expected (*Figure 1—figure supplement 4*). This flattening implied that while the centriole linkages are strong enough to maintain the 9 trip-let microtubules in an ensemble structure, the linkages are also quite flexible and readily deformed when centrioles are isolated from the crowded cellular environment. Furthermore, the centriole refits indicated that other structures, which we had wanted to study, such as the cartwheel, would be non uniformly deformed in our isolated centrosomes. We measured the ellipticity of the A-tubule in our TMT average to have ~10% distortion, in keeping with previously published values for basal bodies and axonemes (*Li et al., 2012*; *Sui and Downing, 2006*). As the TMT appeared not to be affected by the whole-centriole flattening, we focused our efforts on resolving the structure of the TMTs.

Analysis of the TMT global map at a high-contour threshold indicated that apart from the A- and B-tubules, and their respective MIPs, most other features in the map were at less than full occupancy (*Figure 1c*). The protofilaments of the A- and B-tubules were well defined indicating that they repre-sent the core structure of the TMT. The larger seam-stabilizing A-tubule MIP (*Figure 1c* - blue dou-ble arrowheads) was also well defined, while the smaller A-tubule MIP was absent. At this contour level, the A-C linker (*Figure 1c* - green double arrowheads) and pinhead (*Figure 1c* - gold arrow) were completely absent, as was part of the C-tubule. This is consistent with previous work where the pinhead did not persist along the entire length of the centriole (*Paintrand et al., 1992*). Together this was suggestive of structural heterogeneity, especially in the C-tubule.

## Identification of two structural domains along the proximal-distal axis

Initial analysis of whole-population map suggested structural diversity within our sub-volume popula-tion. As a first step, unbiased 3D classification (*Bharat and Scheres, 2016*) suggested two major structural-distinct classes arranged differentially along the centriole length (*Figure 2—figure supple-ment 1*). While the classification was somewhat noisy, likely due to a combination of biological and

computational noise, it indicated changes along the proximal-distal axis, in agreement with previously published data (*Paintrand et al., 1992*). Based on this information, we more systematically grouped together neighboring sets of sub-volumes and made averages from either four 100 nm regions (*Figure 2*) or six 50 nm regions (*Figure 2—figure supplement 2*) along the proximal-distal axis of the centriole. These proximal-distal axis averages showed that the centrioles were formed from two approximately equally-sized domains: a proximal domain, where complete TMTs contained a clear pinhead and A-C linker (*Figure 2a and b*), and a distal domain, where the pinhead, the A-C linker, and part of the C-tubule were missing (*Figure 2c and d*). In the absence of the pinhead, the distal half of the centriole showed a large and unique density emanating from the inner side of the A-tubule (*Figure 2c and d* - gold double arrowheads).

Based on this proximal-distal axis averaging, and the striking differences observed, we regrouped the sub-volumes into two groups, corresponding to the proximal 200 nm and the distal 200 nm of the centriole. These two halves were then aligned independently to generate cohesive proximal and distal averages. Even though each set only contained half the number of subvolumes, the ~3.5 nm resolution was maintained, allowing us to gain additional structural insights into centriole organization (*Figure 3*) (EMD-7776, EMD-7777). As indicated in the earlier averages, the proximal map had a complete triplet TMT decorated with the pinhead (*Figure 3a* - green arrow) and the A-C linker (*Figure 3a* - white double arrowheads). This contrasted with the distal average, where an incomplete triplet lacked any detectable density for the A-C linker (*Figure 3b* - white double arrowheads) or the pinhead. These differences are consistent with previous findings from the Bornens laboratory (*Paintrand et al., 1992*). In addition to the large structural changes, we also noted that while the large A-tubule MIP straddling protofilaments A09 and A10 was equally strong in both proximal and distal averages, the A-tubule MIP on protofilament A11 was absent in the distal centriole (*Figure 3a/d and and b/c*). Together, our proximal and distal averages revealed numerous distinct structural differences.

The most surprising element of the distal-centriole map was the presence of a very large novel L-shaped density that extended from the inner AB-junction, and pointed towards the lumen of the

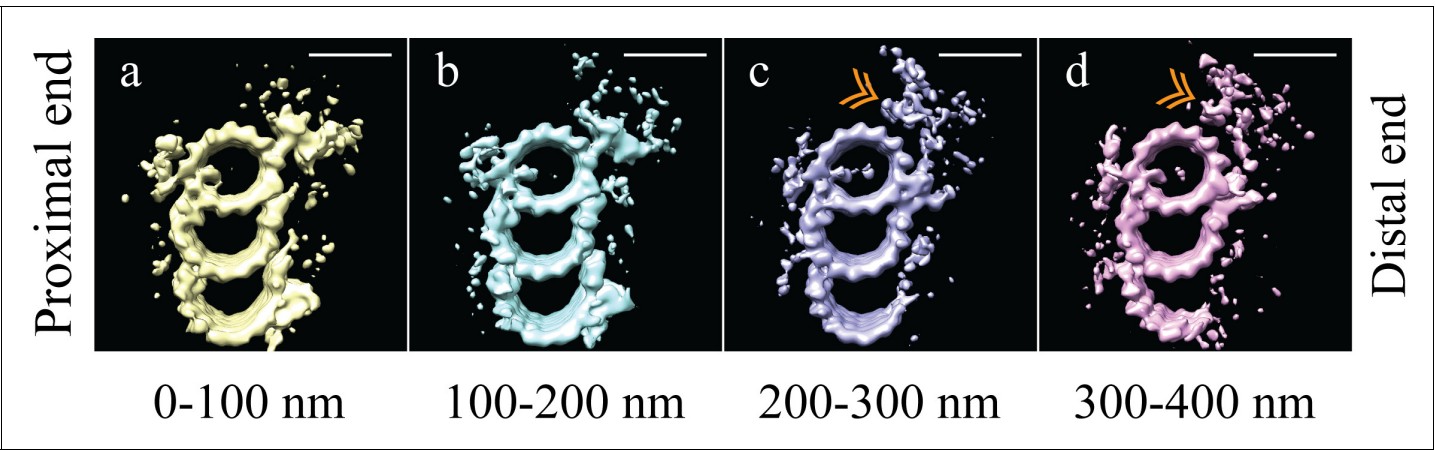

**Figure 2.** The microtubule triplet undergoes significant changes along its proximal-distal axis. The microtubule triplet sub volumes were grouped based on position along the proximal-distal axis of the centriole, and then averaged together to generate maps that represented 100 nm blocks of the centriole. (**a**) and (**b**) There were no major structural differences between the first two groups (0–100 nm and 100–200 nm); complete triplet microtubules were bound by an A-C linker and the pinhead. (**c**) In the third group (200–300 nm) the A-C linker, part of the C-tubule and the pinhead were no longer obvious in the map, but a new density (gold double arrowhead) occupied the space previously occupied by the pinhead. (**d**) In the distal centriole (300–400 nm) a partial, though less complete C-tubule persisted as did the novel density (gold double arrowhead). Scale bars are 25 nm.
DOI: https://doi.org/10.7554/eLife.36851.007

The following figure supplements are available for figure 2:

**Figure supplement 1.** Unbiased classification of the subvolume population suggested centriole domains.
DOI: https://doi.org/10.7554/eLife.36851.008

**Figure supplement 2.** Fine-grain averages along the entire length of the CHO centriole.
DOI: https://doi.org/10.7554/eLife.36851.009

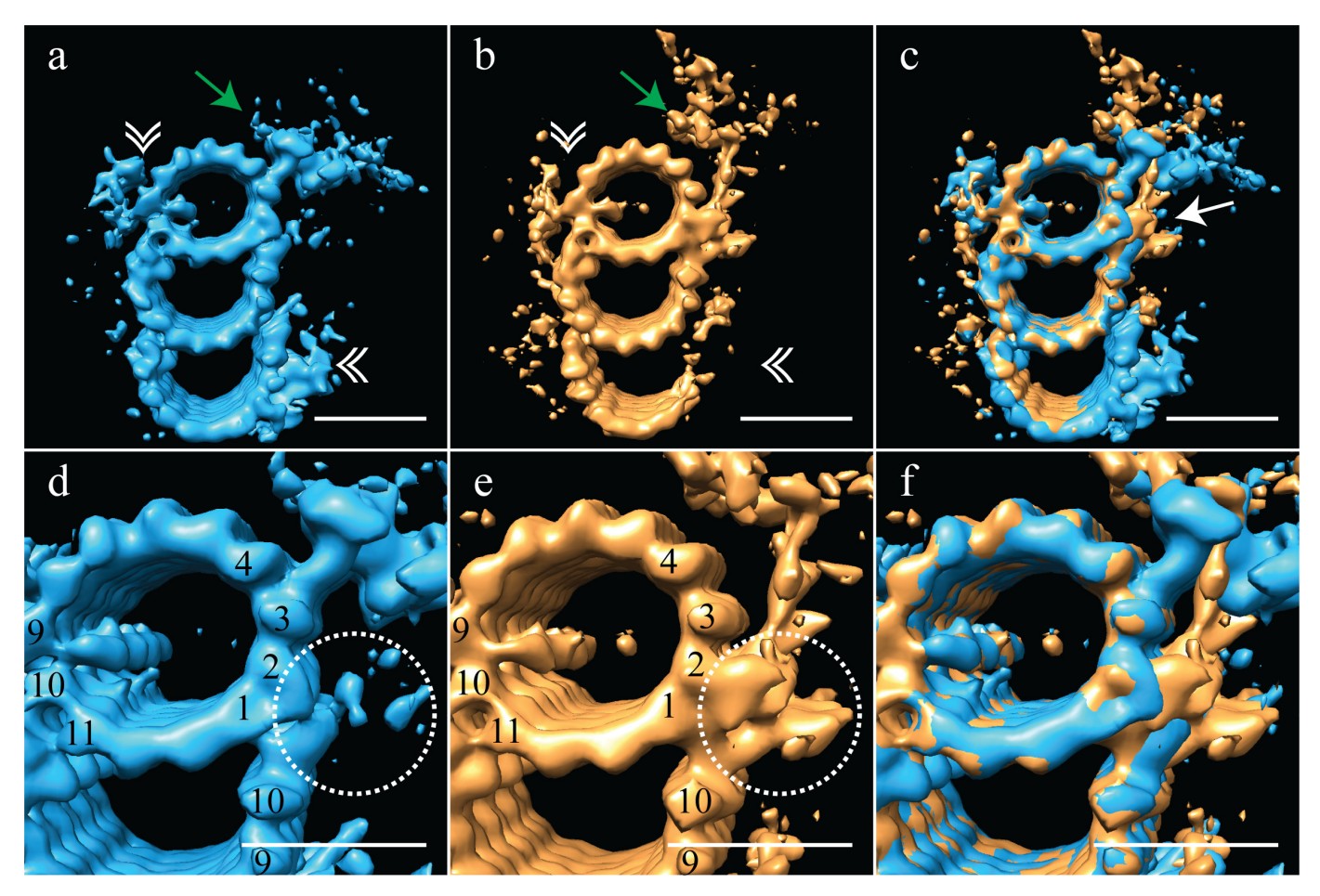

**Figure 3.** The proximal and distal centriole show several major structural differences. (**a**) The proximal centriole was characterized by complete triplet microtubules that showed strong density for both parts of the A-C linker (white double arrowheads) and the pinhead (green arrow). (**b**) In contrast, the distal centriole lacked the A-C linker (white double arrowheads) and instead of a pinhead, the distal map had a density (green arrow) that extended out from the A-tubule. (**c**) A merge of the distal and proximal maps highlighted the differences between them, with differences at the inner AB-junction (white arrow) being the most obvious. (**d**) A zoomed in view of the A-tubule in the proximal centriole showed two MIPs, one straddling protofilaments A09 and A10, the other binding to A11. Note that the inner AB-junction is compact (dashed white circle). (**e**) A zoomed in view of the A-tubule in the distal centriole also showed a MIP straddling protofilaments A09-A10, but the proximal centriole lacked the A11-bound MIP. Here the inner AB-junction was more elaborate and formed the platform from which densities extended (dashed white circle). (**f**) A merged view of both A-tubules allows the differences at the inner AB-junction to be fully appreciated. Scale bars are 25 nm.

DOI: https://doi.org/10.7554/eLife.36851.010

centriole (*Figure 3b* - green arrow). The origin of this L-shaped density was structurally and positionally unrelated to the pinhead, but occupied much the same space, indicating that these structures must be mutually exclusive. An overlay of the proximal and distal maps showed the extent of the differences between the two maps, with the differences at the inner AB-junction being especially prominent (*Figure 3c* - white arrow). The inner AB-junctions showed that while the proximal centriole has a compact junction that mediates A-tubule to B-tubule binding (*Figure 3d* - dashed white circle), the inner AB-junction in the distal centriole was much more elaborate (*Figure 3e* - dashed white circle). In addition to forming the junction between the A- and B-tubules, the distal AB-junction appeared to have additional globular domains binding to protofilaments A01/A02 that acted as a platform from which the L-shaped density extended (*Figure 3e* - dashed white circle). The full extent of the junction differences is apparent in the overlay of the two maps (*Figure 3f*). Thus, the luminal architecture of the distal centriole is strikingly distinct from the canonical pinhead-based structure.

## Mammalian centriole geometry

As discussed earlier, due to the centriole flattening, we were unable to directly determine and apply the symmetry for CHO centrioles. To circumvent the flattening, we used the geometry from previously published data on centrosomes isolated from human lymphoblasts, where centrioles embedded in resin showed no signs of flattening (*Paintrand et al., 1992*). These data also suggested that it is not the isolation procedure that induces flattening, but likely the forces generated during ice-sheet formation. Imposing the published proximal symmetry to our averaged proximal map recreated a 9-fold symmetry that satisfied TMT linkages via the A-C linker (*Figure 4a,c* - orange arrowheads) and the diameter of the ensemble structure was ~235 nm. Applying the published distal symmetry to the distal portion of our CHO centriole produced a map clearly different from the proximal centriole map. The overall diameter of the distal centriole had decreased to ~220 nm, a change achieved by a rotation of the triplets inwards by ~15°. This change in diameter from the proximal centriole accounts for the centriole tapering that we initially observed in the tomograms (*Figure 1a*). Thus, the geometry derived from resin-embedded centrioles can be used to estimate the pre-freezing geometry of our CHO centrioles.

Our distal centriole map (*Figure 4b,d*) elegantly explained many of the structural features that we had observed in our sub-volume averages. The novel densities (*Figure 3b,e*) are now understood to form two connections that link adjacent distal TMTs on both the internal and external centriole surfaces (*Figure 4b* - arrowheads). In particular, the L-shaped density emanating from the inner AB-junction of one triplet zigzagged across and connected to the inner AB-junction of the adjacent triplet (*Figure 4d* - blue arrowhead). On the external surface of the A-tubule, a foot-like appendage extended from protofilament A09 and appeared to bind directly to protofilaments C07/C08 the C-tubule of the adjacent triplet (*Figure 4d* - green arrowhead), generating a non-canonical A-C linker and simultaneously stabilizing the shape and location of the end of the partial C-tubule sheet. Other appendages also extended from the surface of the B-tubule and joined this connection, perhaps adding additional support to this novel linker. Thus, in the absence of the canonical A-C linker in the distal centriole, two new structures and a partial C-tubule mediate the linkage of adjacent triplets, while altering the geometry of the distal centriole to be more axoneme like.

## *Drosophila* centriole structure

After determining the structure of the TMT CHO centriole, we compared it to the DMT structure of the fly centriole. Centrosomes purified from asynchronous *Drosophila melanogaster* S2 cells were frozen on cryoEM grids and cryoET data acquired. Tomograms of the fly centrosomes presented both side-on (28 of 30) and end-on (2 of 30) views of centrioles. In end-on views, it was clear that fly centrioles were composed of predominantly DMT, although SMTs and DMTs could be observed within the same centriole (*Figure 5a*). The SMTs and DMTs were symmetrically arranged around a central hub (*Figure 5a*). *Drosophila* S2 centrioles were much shorter than the CHO centrioles, with an average length of ~175 nm. Intriguingly, end-on centrioles showed brush-like structures, likely representing the core PCM, extending out from the cytoplasmic face of the centrioles and concentrated in the regions between the DMTs (*Figure 5a* – white arrows).To determine the structure of the fly DMTs, disc-shaped sub-volumes, 3 tubulin heterodimers in height were extracted along the length of each DMT rod (N = 30). This gave a dataset of ~3000 unique sub-volumes, which were aligned to a common reference and averaged to generate a global map of the fly DMTs (*Figure 5*) at ~3.5 nm resolution (EMD-7778) (*Figure 1—figure supplement 2*). As expected, this averaging generated a DMT, composed of a 13 protofilament A-tubule and a 10 protofilament B-tubule (*Figure 5b*). The map showed a single prominent MIP bound to the luminal surface of the A09/A10 protofilaments; its shape and position were indistinguishable from the A-tubule MIP that we identified in the CHO map (compare with *Figure 1b*). The spatial and structural conservation of this MIP supports the idea that it plays a highly conserved role. In contrast, the B-tubule MIP bridging the B01 and B02 protofilaments observed in the mammalian centrioles (*Figure 1b*) was absent from the fly B-tubule. Despite this, the fly B-tubule curvature was otherwise indistinguishable from that of the CHO B-tubule, suggesting that a B-tubule MIP is dispensable. Furthermore, the connection between A11 and B01 is maintained in fly centrioles, even in the absence of the B01/B02 MIP, and is perhaps sufficient for B-tubule construction.

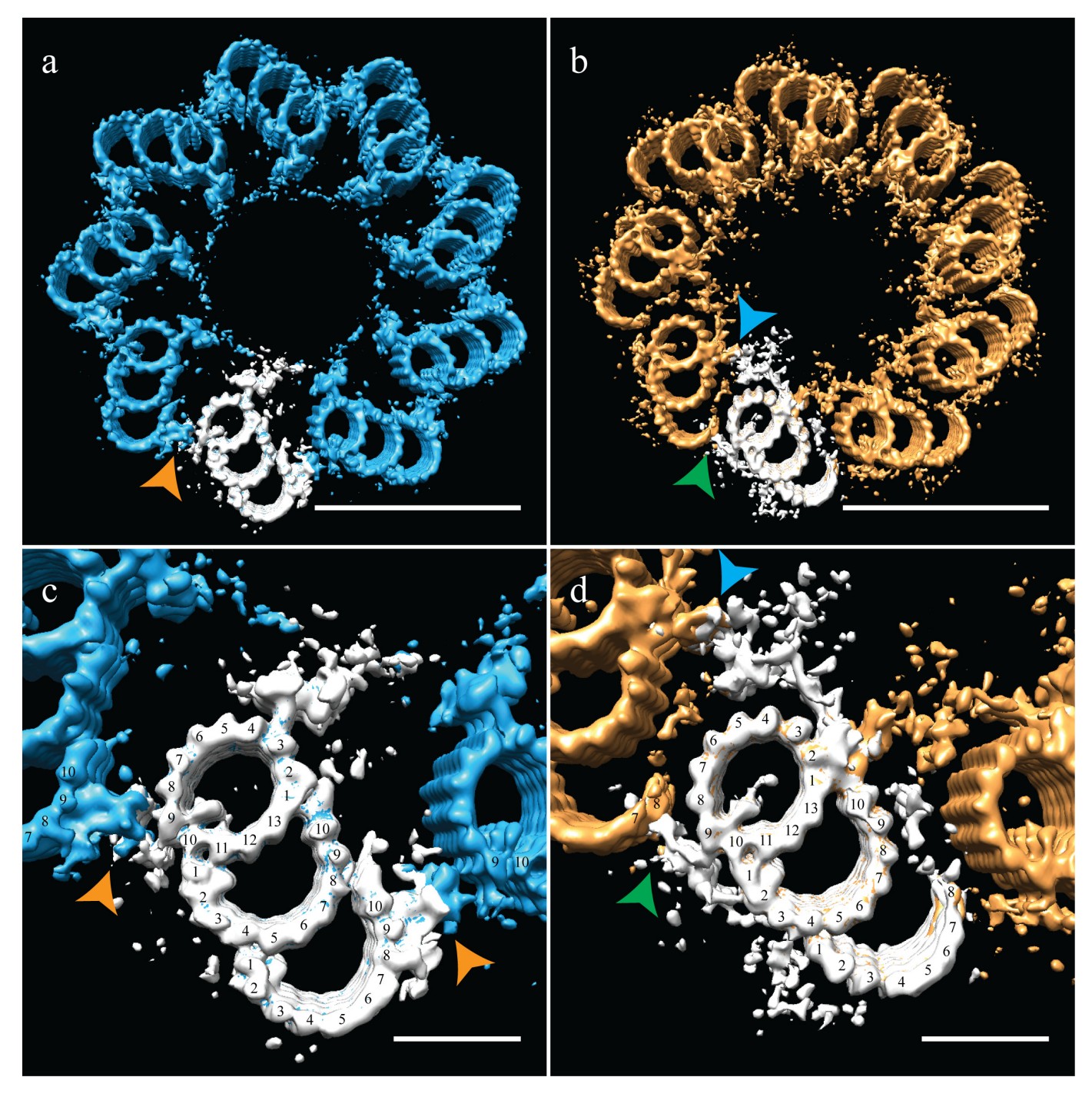

**Figure 4.** Correcting the symmetry revealed novel connections in the distal centriole. Using the symmetry from published studies on mammalian centrioles, the symmetry for the proximal and distal centriole was determined and used to correct for the ice-sheet induced flattening. (a and c) This gave a proximal-centriole geometry where the A-C linkers (orange arrowheads) of adjacent triplet microtubules fit together and recapitulated the geometry as previously published. (b and d) Using the published distal-centriole, geometry showed that adjacent microtubules are connected in two ways, a linker that connected adjacent inner AB-junctions (blue arrowhead), and arm-like densities that connected protofilaments A09 and the persistent C-tubule of the adjacent triplet (green arrowhead). Scale bar is 100 nm in (a) and (b) and 25 nm in (c) and (d).

DOI: https://doi.org/10.7554/eLife.36851.011

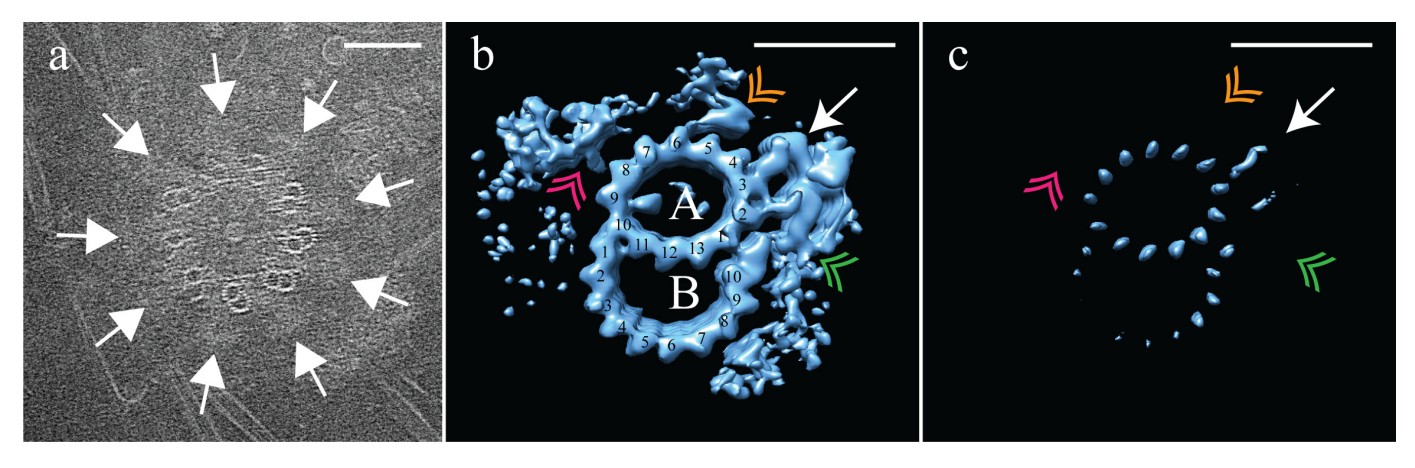

**Figure 5.** Electron cryo-tomography of *Drosophila* melanogaster S2 cells. (a) CryoET showed that fly centrioles were almost exclusively found as single centrioles, and composed of a mixture of singlet and doublet microtubules, symmetrically arranged around a central hub. Amorphous brush-like structures (white arrows) emanated from between adjacent microtubules, possibly representing core PCM. (b) Sub-volume averaging of centriole microtubules generated a microtubule doublet map, with a 13 protofilament A-tubule and a 10 protofilament B-tubule. The pinhead (white arrow) extended from protofilament A03, but was more elaborate than the CHO pinhead, making additional contacts with protofilament A02, and the inner AB-junction (green double arrowhead). The map also showed a novel hinge-shaped density (gold double arrowhead) of unknown function bound to protofilament A06, and a repetitive wishbone-shaped structure (pink double arrowhead) also of unknown function that made weak connections to the A-tubule. (c) Changing the contour level of the map in (b), showed that the A-tubule, the pinhead and a partial B-tubule constituted the core structure, with all other structural elements at less than 100% occupancy. Scale bars are 100 nm in (a), and 25 nm in (b) and (c).

DOI: https://doi.org/10.7554/eLife.36851.012

The outer surface of the fly DMT showed extensive differences from the outer surface of the CHO TMT. A pinhead that extended from the A03 protofilament (*Figure 5b* – white arrow) was much more elaborate than the CHO pinhead, doubling back to make additional contacts with the A02 protofilament and the inner AB-junction (*Figure 5b* – green double arrowhead). In addition to the elaborated pinhead, our DMT maps showed two novel densities that were not observed in the CHO TMTs; a hinge-like density extended from the A05/A06 protofilaments (*Figure 5b* – gold double arrowhead), and a wishbone-shaped structure (*Figure 5b* pink double arrowheads) that made weak connections with protofilament A08.

An analysis of the fly DMT map at high-level threshold indicated that not all structural elements were at 100% occupancy (*Figure 5c*), as was the case for the CHO TMT. Indeed, the only elements that remained at high-level threshold were the A-tubule, pinhead and parts of the B-tubule. The clear presence of the pinhead (*Figure 5c* - white arrow) but lack of an L-shaped density extending from the inner AB-junction, suggests that the fly centriole does not form the distal domain that we identified in the CHO centriole. The heterogeneity of the DMT B-tubule was distinct from the heterogeneity which we had observed in the C-tubule of the CHO TMT. In the case of the fly, DMT B-tubule density was weakest for protofilaments B03-B06, while B01-B02 and B08-B10 showed much stronger density (*Figure 5c*), suggesting differences in density between the B-tubule of the S2 centriole, and the C-tubule of the CHO centriole.

### *Drosophila* centriole geometry

It was upon refitting the DMT map back into the 9-fold symmetry that the reasons for the structural differences between fly and CHO centrioles became clear. As with the CHO centrioles, fly centrioles also showed flattening within the ice sheet. This was the case for all centrioles whose proximal-distal axis was parallel to the ice sheet (28 of 30). However, due to the fact that S2 centrioles are approximately half the length of the CHO centrioles, we imaged two centrioles whose axis was oriented orthogonal to the ice sheet, presenting an end-on view (2 of 30). These end-on centrioles showed no obvious flattening and their geometry closely matched the geometry of the flattening-corrected proximal CHO centriole, with the angles of the DMTs being remarkably similar to those of the CHO

TMTs (*Figure 6—figure supplement 1*). The diameter of S2 centrioles was smaller than CHO centrioles (~210 nm vs. ~235 nm, respectively), a difference that can be accounted for by the absence of the C-tubules.

The reconstructed centriole (*Figure 6a*) revealed that fly DMTs are connected through their A-tubules alone, using a novel A-A linker instead of a modified or elongated A-C linker. This A-A linker consisted of the hinge-like density (similar in shape to the malleus bone of the inner ear) emanating from the A05/A06 protofilaments of one DMT, extending and connecting to the side of the elaborated pinhead of an adjacent DMT (*Figure 6b* - gold double arrowhead).

The 9-fold refit also showed that the wishbone-shaped structure observed in the DMT average (*Figure 5b* - pink double arrowheads), more closely resembled a hollow cone between adjacent DMTs when seen in context (*Figure 6b* - pink double arrowheads). In addition, this density made weak connections back to the A-A linker and the cytoplasmic face of protofilament A08. While this structure is poorly defined and at less than 100% occupancy, its location is consistent with the brush-like structures we observed in the raw data, and likely represents the attachment of core PCM factors to the external surface of the centriole.

## B-tubule biogenesis

As seen in *Figure 6*, the B-tubule does not appear to play a dominant role in DMT-DMT linkages or in the setting or stabilizing centriole geometry. Given the lack of consequence to centriole structure,

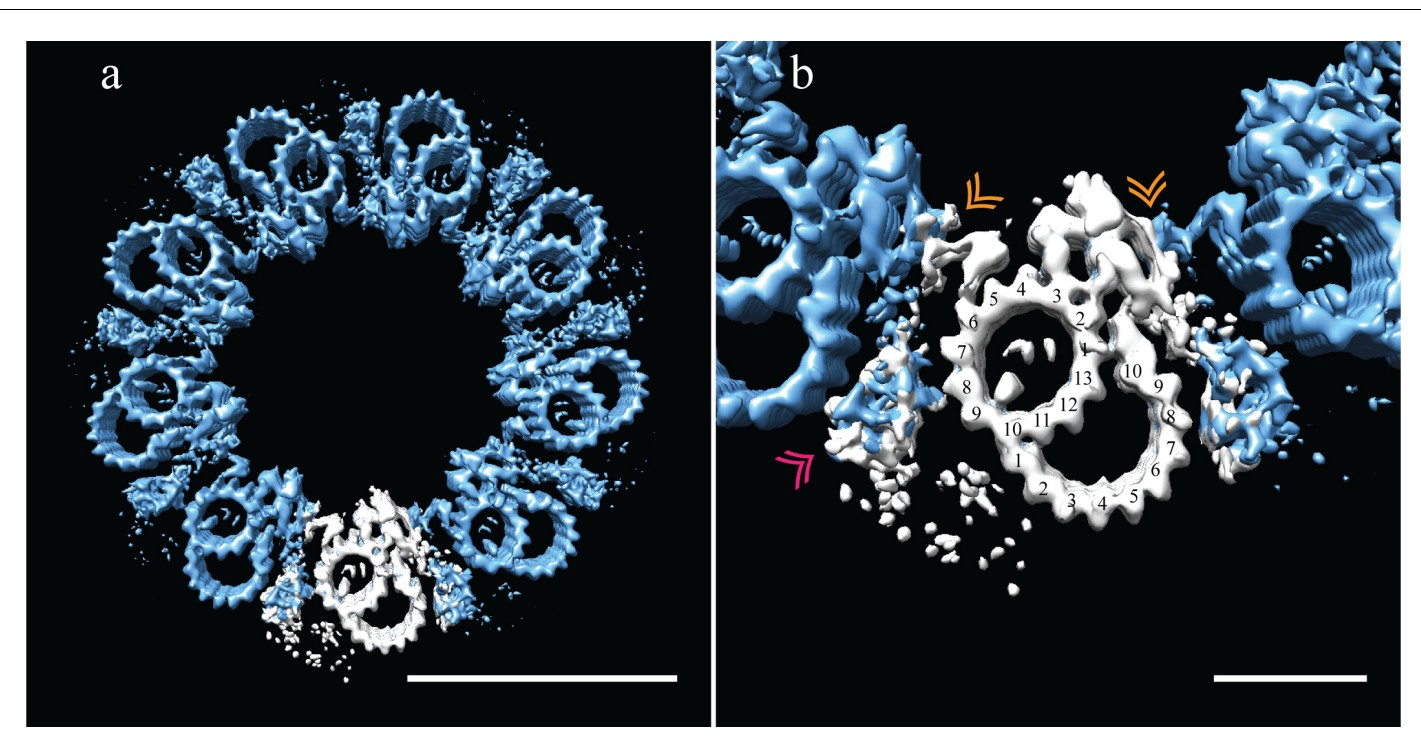

**Figure 6.** Adjacent *Drosophila* melanogaster S2 doublets are connected via a novel A-A linker. (a) The doublet average was fit back into a centriole, which showed no signs of ice-induced flattening, allowing the exact geometry of the S2 centriole to be determined. From this refit it was clear that connections between adjacent doublets were through A-tubule to A-tubule links, hereafter referred to as the A-A linker. (b) A zoomed in view of one doublet in context, showed that the hinge-like structure from one doublet extended and attached to the elaborated pinhead of the adjacent doublet (orange double arrowheads). The second density between adjacent doublets (pink double arrowheads) makes weak connections to both the A-A linker and the A-tubule, and likely represents the brush-like structures that were observed emanating from between doublets in the tomograms. Scale bars are 100 nm in (a), and 25 nm in (b).

DOI: https://doi.org/10.7554/eLife.36851.013

The following figure supplement is available for figure 6:

**Figure supplement 1.** The proximal CHO and *Drosophila* centrioles have the same geometry, but different linkages.
DOI: https://doi.org/10.7554/eLife.36851.014

this would explain why partial or absent B-tubules are well tolerated in the S2 centriole and in the early embryo (*Moritz et al., 1995*; *Callaini et al., 1997*). We reasoned that these B-tubule intermediates in this system might provide insights into B-tubule construction.

Using 3D classification (see Materials and methods), we grouped together all subvolumes that lacked protofilaments B03-B06 and generated a new average (EMD-7779). Even though this was a small number of subvolume (N = 226), there was clear, coherent density for the flanking protofilaments, B01-B02 and B07-B10 (*Figure 7a*). Indeed, when we reexamined the S2 tomograms, we saw that many SMTs were in fact composed of a complete A-tubule with two microtubule stubs attached (*Figure 5a*), corresponding to the average that we generated of the partial B-tubules. The flanking segments of the incomplete B-tubule have approximately the same curvature as the complete B-tubule. Together the data suggest that the partial B-tubules represent a kinetic intermediate in the formation of a complete B tubule. That is, the B-tubule would be constructed by first forming microtubule stubs at each joint on the A-tubule (B01/B02 and B07-B10), subsequently growing together to form a complete B-tubule.

To test this, we examined the distribution of the partial DMTs in the context of a whole centriole. One could imagine scenarios in which the partial DMTs are distributed at the very end of each DMT rod, more akin to a centriole cap, or where full rods randomly transition into partial rods, representing a construction intermediate. To address this issue, we reconstructed a centriole (*Figure 7b*) by fitting back the partial DMT map (pink) to its contributing sub-volumes based on the classification, and then filling in the rest of the gaps with the complete DMT map (blue). The data suggest that entire rods tend to be composed of either complete or partial DMTs, not simply the ends of each DMT rod.

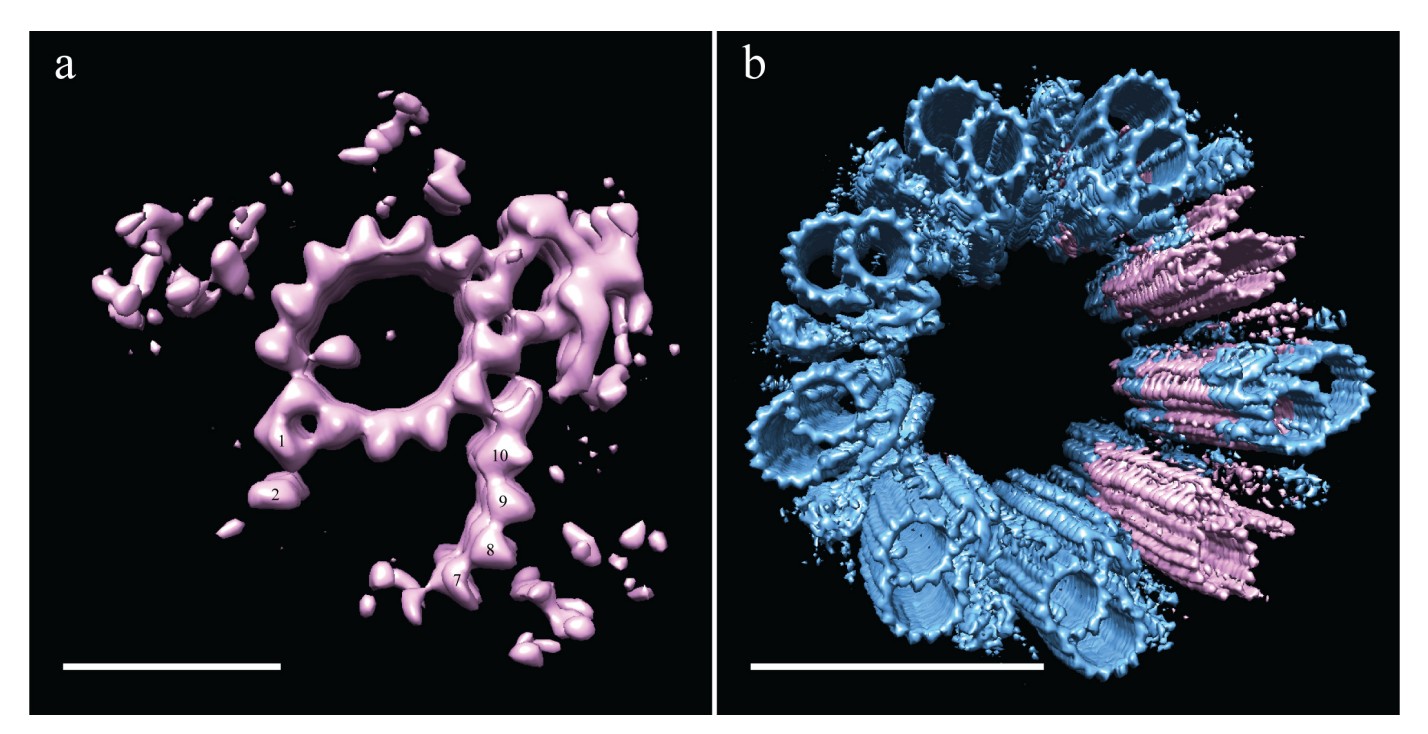

**Figure 7.** Incomplete B-tubules shed light on how B-tubules are constructed in *Drosophila* S2 cells. (a) Starting with the entire dataset (N = 2962), subvolumes with incomplete B-tubules were segregated into one class using classification (see materials and methods) and an average was generated from this subset (N = 226). This average showed strong density for the A-tubule, the pinhead, and parts of the B-tubule. The lack of protofilaments B03-B06 suggested that the B-tubule grows from both sides which then meet to complete B-tubule construction. (b) A centriole was refit to show the positions of the complete and incomplete doublets within the context of the whole centriole, and this suggested that incomplete doublets tend to cluster, both along the proximal-distal axis and around the radial axis of the centriole. Scale bars are 25 nm in (a), and 100 nm in (b).
DOI: https://doi.org/10.7554/eLife.36851.015

## Discussion

Using cryoET of intact centrioles we have, for the first time, generated maps of both mammalian and fly centrioles, providing the highest-resolutions structures to date. These data have provided unexpected insights into how centriolar microtubules are connected in singlet, doublet and triplet microtubules. Far from being universally conserved, our data show distinct strategies used by *Drosophila* and mammals and reveal unexpected heterogeneities within centrioles of each species. Adaptations in the microtubule-microtubule linkages compensate for differences in microtubule number with adjacent doublets in the *Drosophila* centriole connected by an A-A linker, while an A-C linker connects the mammalian triplets. Within the mammalian centriole, there is an abrupt transition between a proximal core domain and a distal domain leading to distinct 9-fold geometries and requiring novel linkages for stabilization in these two distinct zones. The net result is that the distal centriolar segment is much closer in geometry to the ciliary axoneme than is the proximal core.

### Novel linkages maintain the 9-fold symmetry of centrioles despite differences in microtubule structure

The 9-fold symmetry of all motile and nonmotile cilia is universally conserved in all organisms thus far studied. However, the decoration of motile cilia with dyneins and radial spokes differentiates motile cilia from sensory cilia both structurally and functionally. While the 9-fold symmetry of the cilium is highly conserved, the centriolar microtubules that nucleate them are structurally diverse.

Bioinformatic data suggest that the ancestral eukaryotic centriole was composed of triplet microtubules; this architecture has been maintained across the evolutionary spectrum in organisms ranging from trypanosomes to mammals (*Carvalho-Santos et al., 2011*). However, there are numerous examples where centriole microtubule number has changed. In the nematode *Caenorhabditis elegans and Drosophila melanogaster* early embryos, centrioles are composed of SMTs, while centrioles are composed of DMTs in *Drosophila* somatic cells. Previous studies have established that adjacent triplet microtubules in basal bodies are linked through an A-C linker connecting A09 with C08/C09 (*Carvalho-Santos et al., 2011*; *Guichard et al., 2013*; *Li et al., 2012*; *Paintrand et al., 1992*). This is also the case for the proximal half of the mammalian centriole (*Figure 4a*) leaving unexplained how SMTs and DMTs could be linked.

Rather than adapting the A-C linker to fit the doublet centriole, a structurally distinct linkage has evolved to support SMTs and DMTs in *Drosophila* – the A-A linker (*Figure 6a*). A similar connection strategy may very well be employed in other organisms without TMTs, such as *C. elegans*. From our maps, this A-A linker consists of a hinge-like density extending from protofilaments A05/A06 and reaching across to make attachments to a new domain of an elaborated pinhead from the adjacent doublet. Structurally, the A-A linker does not resemble the A-C linker. This is perhaps unsurprising given that the A-A linker resides in a different location from the A-C linker. Moreover, this linkage also appears to serve as the primary anchor point for the surrounding PCM (*Figures 5a* and *6b*) connecting either directly or indirectly to proteins such as Asl and Plp (*Fu and Glover, 2012*; *Mennella et al., 2012*; *Roque et al., 2018*). Given the diversity in the linkages, yet the maintenance of the geometry, it is clear that evolution puts a premium on maintaining the geometry of the centriole, an organization that is instructive in determining cilium geometry.

### Centriole domains

Rather than being homogeneous, the proximal and distal portions of the mammalian centriole have distinct architectures. The proximal ~200 nm are decorated with the pinhead, consistent with previous structures of basal bodies (*Guichard et al., 2013*; *Li et al., 2012*), while the ~200 nm distal domain lacks the pinhead and exhibits a distinct architecture. In the fly, only the ~175 nm, pinhead-decorated domain is present (*Figure 5*). These data suggest that all centrioles share a common core,~200 nm in length, but that mammalian centrioles are distally extended. Since fly DMT centrioles are only competent to form sensory cilia, whereas mammalian TMT centrioles can form both motile and sensory cilia, it is tempting to speculate that the distal extension endows the centriole with this expanded potential.

The distal mammalian centriole transitions to a narrower diameter and a geometry more reminiscent of an axoneme. This change in geometry parallels the appearance of two novel linkers (*Figure 4*). One is a non-canonical A-C linker that links protofilament A09 with protofilaments C07/C08

at the end of the partial C-tubule, helping to explain the persistence of the partial C-tubule along the distal centriole. Whether the incomplete C-tubule helps in the formation of the linker or the linker blocks C-tubule completion cannot be discerned from our data. The second distal linker zig-zags from one inner AB-junction to an adjacent inner AB-junction. In the absence of a canonical A-C linker in the distal centriole, these novel linkers work together to link adjacent TMTs and enforce more axoneme-like and perhaps more rigid geometry.

The mechanism by which the pinhead terminates at ~200 nm along the centriole from the proximal end is unclear, but its disappearance along the centriole axis happens quite abruptly indicating that it is precisely determined, rather than developed statistically. One possibility is that length determination resides within the cartwheel rather than within the centriole, with cartwheel length being actively controlled, thereby defining the length of the pinhead-containing region. Indeed, recent work in the *Drosophila* embryo shows that cartwheel length is actively regulated (*Aydogan et al., 2018*).

## MIPs

A pronounced feature of previously determined basal body and axoneme structures is the presence of a plethora of MIPs that decorate the luminal surface of the microtubules (*Ichikawa et al., 2017*; *Li et al., 2012*; *Linck et al., 2014*; *Maheshwari et al., 2015*; *Oda et al., 2014*). Due to their position and structure, it is believed that many of these MIPs add to the structural integrity of the microtubules, especially important within axonemes that are generating or subject to force.

While the purpose of the A-tubule MIP that straddles protofilaments A09 and A10 in both fly and mammalian centrioles is unknown (*Figures 1b* and *5b*), its common location and shape are suggestive of a conserved role in microtubule stability. A similarly located density has been identified in axonemes, and is called MIP2. We favor the idea that this MIP is binding across the seam of the A-tubule, the position of which has been assigned from recent cryoEM work in axonemes (*Ichikawa et al., 2017*). Consistent with this, these two protofilaments are more widely separated than the others and the MIP binds diagonally across the gap, thus preferentially interacting with the same tubulin type (either α-tubulin or β-tubulin) on either side of the seam (*Figure 1*). A MIP straddling the microtubule seam may confer extra stability to the A-tubule, accounting for the extraordinary stability of centriolar microtubules. The larger spacing between protofilaments A09 and A10 (*Figure 1c*), may also allow the A-tubule to distort creating unique binding sites on the A-tubule. Consistent with this idea, protofilament B01 binds to the lateral surface of protofilament A10, suggesting that an A10-B01 lateral interaction may contribute to B-tubule nucleation or stabilization. Furthermore, both the canonical- and non-canonical A-C linkers bind to protofilament A09 (*Figure 4b and d*). Taken together, we believe that this seam-MIP combination is a hotspot providing unique sites that contribute to A-tubule functionality.

Based on their binding location, we initially speculated that the mammalian B- and C-tubule MIPs also played some role in tubule formation or stabilization. While the MIP straddling A09 and A10 in both species supports a role in B-tubule initiation, the MIP linking B01/B02 does not as it is only present in the mammalian TMT. Moreover the B-tubules in both species have the same curvature and protofilament number independent of this B01/B02 MIP. Given the apparent pause in B-tubule completion observed in the S2 centriole, it seems more plausible that a B01/B02 MIP enhances construction kinetics of the B-tubule, ensuring its timely construction. The corresponding and similarly shaped MIP at C01/C02 is likely to play an analogous role in facilitating efficient initiation and timely completion of the C-tubule which is necessary for formation of a stable A-C linker. The A-A linker of the fly centriole imposes no such restraint on B-tubule construction, allowing the completion of the B-tubule to be somewhat *ad hoc*. The similar shape and parallel roles of the B and C tubule MIPS also suggests that they likely share molecular components. An intriguing possibility is that this Y-shaped MIP corresponds to the recently discovered TED complex composed of delta and epsilon tubulins and two accessory proteins (*Breslow et al., 2018*; *Wang et al., 2017*). In the absence of the delta- and epsilon-tubulin complex, centrioles do form, but are unstable and subsequently disintegrate, suggesting that the delta/epsilon-tubulin complex is necessary to stabilize the TMT. Consistent with this, these proteins are absent in *Drosophila*, and S2 centrioles lack those MIPs.

## Biogenesis of the B-tubule

Incomplete B-tubules in the fly centriole have previously been described and are thought to represent construction intermediates in the rapidly dividing embryo (*Callaini et al., 1997*; *Moritz et al., 1995*). Intriguingly, classification of the B-tubule gave a class in which microtubule sheets curve inward from both the inner- and outer-AB junctions (*Figure 7a*). Surprisingly, these 'microtubule stubs' are sufficiently stable to have a curvature indistinguishable from the complete tubule (*Figure 7a*). Based on these microtubule stubs, we favor the idea that we are capturing B-tubule construction intermediates. If this is indeed the case, it implies that B-tubule construction is bidirectional, with the sheets presumably meeting to complete the B-tubule.

We do not see any such B-tubule, microtubule stubs in our CHO centrioles as all those B-tubules are complete, and so we have no hints as to how B-tubules form in CHO centrioles. Microtubule stubs reminiscent of those we see in our cryo-tomography data of S2 centrioles have previously been observed in thermally fractionated (40° C for 5 min) Sea Urchin DMTs (*Linck et al., 2014*). While we did not subject our centrioles to thermal fractionating, we cannot exclude the possibility that the microtubule stubs seen in our S2 data represent damaged or depolymerizing B-tubules that occurs during purification or grid preparation.

While our cryoET data have resolved structures of the fly and mammalian centriole for the first time, it has only scratched the surface on determining the molecular structure of the centriole. Of relevance to many human diseases is how the centriole changes during ciliogenesis to become a membrane-anchored basal body, and allow for the establishment of structures like the transition zone and ciliary necklace, structures that are central to cilium formation and function. Many, if not most, of the densities that we see in centrioles are molecularly unidentified and of unknown function. With so many intriguing yet unidentified densities being revealed by this study and others, the focus now shifts to assigning their molecular identities and determining their functional roles.

## Materials and methods

**Key resources table**

| Reagent type | Designation | Source/Reference | Identifiers |
|---|---|---|---|
| Cell Line (Cricetulus griseus) | Chinese Hamster Ovary/CHO | UCSF Cell Culture Facility | RRID:CVCL_0214 |
| Cell line (Drosophila melanogaster) | Drosophila S2U/S2 | Vale laboratory | Flybase ID: FBtc9000006 |
| Chemical compound/drug | Demecolcine | Sigma Aldrich | SKU: D7385 |
| Chemical compound/drug | Nocodazole | Sigma Aldrich | SKU: M1404 |
| Software | Relion | *Bharat and Scheres, 2016* | PMC5215819 |
| Commercial assay or kit | Mycoplasma testing kit | Lonza | LT07-318 |

### Centrosome purification

Centrosome purification protocols were based on published protocols (*Reber, 2011*). CHO cells (purchased from the UCSF Cell culture facility) were maintained at 37 °C (in D-MEM medium with 10% fetal bovine serum and 1x Penicillin-Streptomycin-Glutamine), and *Drosophila* S2 (Vale laboratory stock FlybaseID: FBtc9000006) cells at 25 °C (in Schneider's medium, with 10% fetal bovine serum and 1x Antibiotic-Antimycotic solution). All tissue-culture solutions were from Gibco/Thermo Fisher, flasks were from Corning, and all chemicals from Sigma-Aldrich. Cell lines were tested periodically to confirm that stocks remained mycoplasm-free (Lonza MycoAlert Mycoplasma Detection Kit LT07-318).

Cells from $4 \times 225$ cm$^3$ flasks were resuspended and reseeded on $6 \times 225$ cm$^3$ flasks the night before the centrosome prep. On the morning of the prep, the CHO medium was replaced with fresh CHO medium containing Nocodazole at 10 µg/mL, and the S2 cell medium with fresh S2 medium containing Colcemid at 5 µm/mL. The flasks were then replaced to their incubators from 90 min.

In a 4 °C coldroom, the medium was poured off and the cells washed quickly but gently with 1x TBS (50 mM Tris-Cl, pH 7.6; 150 mM NaCl), 8% sucrose in 0.1x TBS, 8% sucrose in water, and finally in a pre-lysis wash buffer (1 mM K-HEPES pH 7.2, 0.5 M MgCl$_2$, 0.1% 2ME). 6 mL of lysis buffer (1

mM K-HEPES pH 7.2, 0.5 M MgCl$_2$, 1% 2ME, 2% NP-40) was added to each flask, and the flasks were then rocked gently for ~20 min. The cloudy lysate (~40 mL) was collected in a 50 mL conical tube, with care taken to leave the cell carcasses attached to the bottom of each flask, and the lysate made to 10 mM K-PIPES pH 7.2. The lysate was then clarified of cell nuclei and contaminants by spinning at 1500 x g for 10 min at 4 ˚C.

This clarified lysate was then layered on top of a 2 mL sucrose cushion (60% w/w in gradient buffer - 10 mM K-PIPES pH 7.2, 0.1% NP-40, 500 µl of 0.1% 2ME). Spinning for 30 min in a JS13.1 rotor at 24,000 x g at 4 ˚C concentrated the centrosomes in a white layer on top of the sucrose cushion. After the spin the vast majority of the supernatant was aspirated, leaving behind ~2 mL of the sucrose pad and ~1.5 mL of concentrated centrosomes in a puffy white layer. These layers were thoroughly mixed to give ~3.5 ml of ~35% sucrose. This solution was layered on top of a two-step sucrose gradient in an SW60 centrifuge tube; 250 µL of 70% and 250 µL 55% in gradient buffer. This miniature gradient was spun in an SW60 rotor at 55,000 rpm for 30 min at 4 ˚C, yielding a white layer of centrosomes at the interface of the 70% and 55% sucrose layers. The centrosomes were then collected by manual fractionation using a pipette, and aliquots snap frozen in liquid nitrogen.

Prep success and contamination levels were evaluated using a microtubule-regrowth assay, followed by centrosome/microtubule staining and light microscopy analysis. On the electron-microscopy grid the contrast was low and it was impossible to differentiate the S2 centrosomes from dirt. We overcame this problem by regrowing centrosome microtubules, enabling us to follow microtubules and to distinguish centrosomes from dirt.

## Grid preparation

Electron-microscopy grids (Quantifoil R 2/2 Cu 200 mesh) were glow discharged for 25 s, and then placed on top of an ice-chilled aluminum block. 3 µL of a centrosome solution (diluted centrosomes with 3 µM bovine tubulin, 1x BRB80, 1 mM GTP and a dilute 10 nm fiducial-gold solution) was placed on each grid, and centrosomes allowed to settle on the grid for ~15 min. A grid was then picked up with tweezers and loaded into the chamber of an FEI Vitrobot, pre equilibrated to 100% relative humidity and 30 ˚C. After allowing 1 min of microtubule regrowth from centrosomes, the grids were blotted and plunge frozen in liquid ethane.

## Image acquisition and processing

Imaging was performed on an FEI Tecnai Polara microscope operating at 300 kV, controlled using our in-house software, UCSF tomography. Images were acquired on a Gatan K2 camera, through a Gatan GIF Quantum Energy Filter, at a dose rate of ~8 e/px/s and a total dose of ~80 e/A2. For each centrosome a tilt series was acquired from +60˚ to −60˚ with 1˚ angular sampling. Tilt series were aligned using gold-bead alignment in IMOD (University of Colorado) and tomograms generated using the EWBR-method in Prism (*Chen et al., 1996*). Datasets where high tilts were obscured by a grid bar or aperture, and datasets where gold-bead alignment was unsuccessful, were excluded from the data analysis.

Subsequent sub-volume alignment, averaging, CTF correction and classification were performed using RELION (*Bharat and Scheres, 2016*). Some manual inspection and curation of the data was performed to check for consistency of angles and translations along and between rods of a given centriole. 5-heterodimer long subvolumes were aligned to a reference model to determine the periodicities of the microtubule-binding densities (*Figure 1—figure supplement 1*). All non-tubulin densities that were identified were contained within a 3-heterodimer/24 nm segment, and this was used for all data alignment and analysis.

For 3D classification, a tight mask was generated around the region of interest (the pinhead in the CHO average of protofilaments B03-B06) and 100 rounds of classification were performed without alignment to generate stable classes containing or missing the region of interest.

UCSF Chimera (*Pettersen et al., 2004*) Imagej and Adobe Creative Cloud were used for data analysis and figure preparation.

## Acknowledgments

We wish to thank members of the Agard and Vale laboratories, and the cryoEM community at U.C. S.F for invaluable support and scientific discussions. This work was made possible by generous funding from NIH grants GM031627 (D.A.A.), GM118099 (D.A.A.) and GM118106 (R.D.V.), and by funding from HHMI (D.A.A. and R.D.V.).

## Additional information

### Funding

| Funder | Grant reference number | Author |
| --- | --- | --- |
| Howard Hughes Medical Institute | | Ronald D Vale Dr David A Agard |
| National Institute of General Medical Sciences | GM031627 | David A Agard |
| National Institute of General Medical Sciences | GM118099 | David A Agard |
| National Institute of General Medical Sciences | GM118106 | Ronald D Vale Dr |

The funders had no role in study design, data collection and interpretation, or the decision to submit the work for publication.

### Author contributions

Garrett A Greenan, Conceptualization, Data curation, Software, Supervision, Validation, Investigation, Visualization, Methodology, Writing—original draft, Project administration; Bettina Keszthelyi, Data curation, Software, Methodology; Ronald D Vale, Conceptualization, Resources, Funding acquisition, Investigation, Methodology, Project administration, Writing—review and editing; David A Agard, Conceptualization, Resources, Supervision, Funding acquisition, Investigation, Methodology, Project administration, Writing—review and editing

### Author ORCIDs

Garrett A Greenan http://orcid.org/0000-0002-9045-7666
Ronald D Vale https://orcid.org/0000-0003-3460-2758
David A Agard http://orcid.org/0000-0003-3512-695X

### Decision letter and Author response

Decision letter https://doi.org/10.7554/eLife.36851.028
Author response https://doi.org/10.7554/eLife.36851.029

## Additional files

### Supplementary files

• Transparent reporting form
DOI: https://doi.org/10.7554/eLife.36851.016

### Data availability

Average maps have been submitted to the Electron Microscopy Database (EMDB) (accession nos EMD-7775, EMD-7776, EMD-7777, EMD-7778, EMD-7779)

The following datasets were generated:

| Author(s) | Year | Dataset title | Dataset URL | Database, license, and accessibility information |
| --- | --- | --- | --- | --- |
| Greenan GA, Keszthelyi B, Vale | 2018 | Whole-population, triplet-microtubule map from a Chinese | http://www.ebi.ac.uk/pdbe/entry/emdb/EMD- | Publicly available at the Electron |

| | | | | |
|---|---|---|---|---|
| RD, Agard DA | | hamster ovary (CHO) centriole | 7775 | Microscopy Data Bank (accession no. EMD-7775) |
| Greenan GA, Keszthelyi B, Vale RD, Agard DA | 2018 | Proximal domain, triplet-microtubule map from a Chinese hamster ovary (CHO) centriole | http://www.ebi.ac.uk/pdbe/entry/emdb/EMD-7776 | Publicly available at the Electron Microscopy Data Bank (accession no. EMD-7776) |
| Greenan GA, Keszthelyi B, Vale RD, Agard DA | 2018 | Distal domain, triplet-microtubule map from a Chinese hamster ovary (CHO) centriole | http://www.ebi.ac.uk/pdbe/entry/emdb/EMD-7777 | Publicly available at the Electron Microscopy Data Bank (accession no. EMD-7777) |
| Greenan GA, Keszthelyi B, Vale RD, Agard DA | 2018 | Whole-population, doublet-microtubule map from a Drosophila melanogaster S2 cell centriole | http://www.ebi.ac.uk/pdbe/entry/emdb/EMD-7778 | Publicly available at the Electron Microscopy Data Bank (accession no. EMD-7778) |
| Greenan GA, Keszthelyi B, Vale RD, Agard DA | 2018 | Partially-constructed, doublet-microtubule map from a Drosophila melanogaster S2 cell centriole | http://www.ebi.ac.uk/pdbe/entry/emdb/EMD-7779 | Publicly available at the Electron Microscopy Data Bank (accession no. EMD-7779) |

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
