## [Decision Letter]

Thank you for submitting your article "Insights into centriole biogenesis and evolution revealed by cryoTomography of doublet and triplet centrioles" for consideration by *eLife*. Your article has been reviewed by three reviewers (including myself, Reviewer #1), and the evaluation has been overseen by me, Jordan Raff, as Guest Reviewing Editor and Anna Akhmanova as the Senior Editor. The following individual involved in review of your submission has agreed to reveal his identity: Tim Stearns (Reviewer #3).

I have discussed the reviews with the other reviewers and I am pleased to say that we think your article is of sufficient interest to warrant publication in *eLife*, although there are a number of issues that will need to be addressed prior to acceptance. I discuss the major points below and also provide lightly edited versions of the major points raised by the other two reviewers in the hope that these comments will help you address the four Major points listed below.

Major Points:

1) You will need to supply more information about how the data was analysed, and why certain strategies were chosen. Why, for example, was a 24nm periodicity chosen as the basis for your averaging strategy? Were you able to measure any longitudinal periodicities from your data? Did you combine mother and daughter centrioles in your averaging? In addition, you claim a resolution of 25 Å, but give no details as to how this was determined and whether this applies to the whole structure.

2) You will need to explain more clearly why certain aspects of these structures were not analysed. Mammalian centrioles are generally thought to have a distal region composed of DMTs: was this absent from the centrioles you analysed here, or was it ignored for technical reasons? Similarly, an analysis of cartwheel structure would be extremely useful to the field, yet you make no attempt to do this, even in fly centrioles where most centrioles have this structure (and it is clearly evident in the centriole shown in Figure 5A).

3) Perhaps these omissions are due to the "flattening" problem but, if so, this needs to be better explained. Indeed, we felt it might be better to more prominently discuss the "flattening" problem at the start of the Results section, and to more directly describe how this influenced your subsequent analysis.

4) It would have greatly increased the impact of this work if you had examined the structure of fly centrioles lacking pericentrin/PLP. Your previous work showed that PLP is a key component of the interphase centriole in flies, and very likely a component of the amorphous brush-like structures you highlight in Figure 5A (Mennella et al., 2012 – see also Roque et al., 2018, which should be mentioned). We would have thought this analysis would be relatively straightforward. We would not insist on this but, if you really can't get this data, it would be helpful to discuss any insights you have as to why this approach may not be as straightforward as one might assume.

Reviewer comments informing summary review:

*Reviewer #2*:

1) The basis of averaging strategy, such as nine-fold symmetry and 24nm (3 tubulin dimers) periodicity, which is the fundamental basis of this study, is not clearly indicated. In the case of the axoneme, there are a number of studies (such as Goodenough and Heuser or by Peter Satir in 1980s) which proved 96nm repeat. In the axoneme, some structures such as MIPs follow, not 24nm, but other (16nm, 32nm) periodicity. 9fold symmetry and symmetry breaking (Hoops & Witman; Bui et al., 2012, JCB) have been described in the axoneme field. Are there any evidence that these centrioles follow 24nm periodic and 9fold symmetry? Alternative periodicity cannot be excluded to explain some weak densities. In the tomogram of vertically oriented *Drosophila* centrioles, it will be possible to discuss 9fold symmetry. Did they mix daughter and mother centrioles?

2) Plunge-frozen centrioles used in this study were (except the one oriented vertical in ice) flattened. The authors fit averaged subtomograms from these tomograms but fitted it to another 3D tomogram which retained 9fold in the past work. Based on this fitting they discussed the positions of the A-C and A-A linkers. If these linkers keep linking the same protofilaments, they should be oriented differently in the flattened centrioles and thus should be smeared out by subtomogram averaging. What do the authors think? Do the linkers actually detach from the adjacent triplet (or doublet), retaining the angles of protrusion?

3) Classification process is not clear. In the text, they stated that four separate structures were obtained, but only two are shown in Figure 2—figure supplement 1.

Figure 2—figure supplement 1C is interesting. Do all the mammalian centrioles, which this reviewer thinks to have variety of the entire length, share the same length of the proximal region? How do they explain significant amount of "red" subtomograms within the white area in Figure 2—figure supplement 1C? Do they appear in all the centrioles analyzed in this work?

Reviewer #3:

1) Doublet microtubule region of the centriole: The authors extracted the triplet microtubule region of centrioles for subvolume averaging (subsection “Mammalian centriole structure”). However, it is well-known that mature (as these seem to be) mammalian centrioles have a distal region composed of doublet microtubules (Paintrand et al., 1992). Analysis of this region is especially important to understanding the transition between the centriole and ciliary axoneme. The authors do describe differences that appear between the proximal end of the centriole and the region where the C-tubule becomes only a partial tubule, but it would helpful to know whether they averaged the doublet microtubule region proper, to determine whether there may be similarities or differences between the doublet and triplet microtubule regions. If not, this should be clearly stated in the text, in addition to a rationale about why this region was not analyzed. This is potentially very important in understanding whether the mature mammalian centriole could more properly be thought of as a core centriole with an axoneme initial segment on its distal end.

2) The cartwheel: The authors state that due to the centriole cycle, it is likely that none of the CHO cell centrioles retained their cartwheels. However, only centrioles in G1 lack cartwheels – procentrioles in S-phase, G2, and even early mitosis have cartwheels. In addition, centriole pairs in which the mother and procentriole are orthogonal and close to each other are probably engaged and thus be at a stage prior to removal of the cartwheel after disengagement (Shukla et al., 2015, Nature Communications). Thus, the centriole pair in Figure 1A is likely engaged based on orientation and proximity of the two centrioles. Did the authors analyze the base of this daughter centriole (or procentriole, if still engaged) to determine whether there is a cartwheel present? Were there other orthogonal pairs in their dataset that may have cartwheels in the procentrioles? This would be especially important to understand the nature of centriole engagement, for which no structural correlates are known. If no cartwheel was found in the orthogonal centriole pair(s), the authors should note that there is a possibility that the cartwheels were lost from CHO cell centrioles during isolation and fixation. *Drosophila* centrioles, in contrast to mammalian centrioles, retain their cartwheel though the cell cycle, so one would expect that all fly centrioles examined have, or had, cartwheels. Consistent with this, the fly centriole shown in Figure 5A has a cartwheel, based on the presence of a central tubule, which is likely the hub of the cartwheel. Was the cartwheel retained in all fly centrioles examined? If not, it would be important to note that the preparation results in loss of some cartwheel structure. This might also allow a determination as to whether it is retention of the cartwheel that better allowed some *Drosophila* centrioles to retain their structure and overall geometry.

3) Evolution: The authors previously published structural analysis of *Chlamydomonas* centrioles, which, like mammalian centrioles, have triplet microtubules. To maximize the evolution impact of this work, it would be helpful to know whether the linkages that reinforce 9-fold symmetry are similar between *Chlamydomonas* and mammalian centrioles, as well as whether any of the non-microtubule densities observed in C.r. are similar to those described here. In addition, the authors note that the density spanning A09 to A10 is conserved between all centriole and axoneme structures – could they state more definitively the degree to which the shape of this structure is similar between species, and thus whether it is consistent with a similar protein (complex) forming this structure?

4) Partial B-tubules: The authors interpret the presence of partial B-tubules that lack protofilaments B03-B06 as evidence that B-tubule assembly in *Drosophila* occurs from both the inner and outer edges of the A-tubule. However, it is known that during thermal fractionation of axonemes, B-tubule stubs attached to the A-tubule remain, while the rest of the B-tubule is lost (Linck and Langevin, 1981, JCB; Linck et al., 2014). Can the authors rule out the possibility that the partial B-tubule is a degradation product? If not, this interpretation should either be removed or qualified accordingly.

---

## [Author Response]

Major Points:1) You will need to supply more information about how the data was analysed, and why certain strategies were chosen. Why, for example, was a 24nm periodicity chosen as the basis for your averaging strategy? Were you able to measure any longitudinal periodicities from your data? Did you combine mother and daughter centrioles in your averaging? In addition, you claim a resolution of 25 Å, but give no details as to how this was determined and whether this applies to the whole structure.

Periodicity and sub-tomogram size.

To be clear the 24 nm was the relevant longitudinal segment size, and doesn’t imply any underlying pattern of periodicity. Normally one makes this segment size as small as is consistent with signal to noise and alignability in an effort to minimize incoherence from bending, etc. Typically underlying periodicities are some multiple of the basic 4 nm tubulin repeat. Prompted by the reviewer’s question, we did assess whether we were missing a repeat larger than 8 nm (the largest observed) and generated a new figure (Figure 1—figure supplement 1). For this, we generated a starting model filtered to 10 nm resolution and devoid of repeating units such as the pinhead, A-C linker, or A09-A10 MIP. Furthermore, the proximal and distal subvolumes were taken as two separate classes, and independently aligned to this starting model.

Using subvolumes that were 40 nm in height (5 tubulin heterodimers), we did a translational search ( ± 13 nm) against the filtered model. After several iterations of searching, there was little change, and we present the averages after 10 iterations (Figure 1—figure supplement 1). This experiment showed the 8 nm periodicity of the A09-A10 MIP to be the same in the two independent, proximal and distal averages. The proximal average showed the pinhead and A-C linker with an 8 nm periodicity, while the distal average showed the novel inner-AB junction to also have 8 nm periodicity. We believe this is the best way to test the possibility of other periodicities along the longitudinal axis.

Mother versus daughter centrioles

Of the 9 centrioles that we used to generate the CHO TMT map, only one of the centrioles is unambiguously a daughter centriole (Figure 1). All other centrioles are not part of an obvious, orthogonal pair. Furthermore, we see no evidence of distal- or subdistal-appendages in any of the centrioles that we imaged, even in the centriole that is unambiguously a mother centriole (Figure 1). We have added a sentence addressing the appendages to the main text of the paper (Results first paragraph). Without obvious appendages, and in the absence of orthogonally arranged centrioles, we had no reliable way to determine whether non-paired centrioles are mother or daughter centrioles in our EM data, and so we averaged all centrioles together.

In future projects we would like to acquire a lot more data of orthogonally-arranged centriole pairs and be able to generate separate maps of mother and daughter centrioles. Such studies will help to address pertinent questions about mother versus daughter centrioles as raised by the reviewers.

Resolution

In the original manuscript, we reported a resolution of 2.5 nm for our maps. This calculation was based on the signal to noise ratio using Class3D alignment in Relion. A real-space local resolution assessment using ResMap also indicates the bulk of the triplet as being between 2.4 nm and 2.8 nm (Figure 1—figure supplement 3). A more unbiased method that avoids the possibility of over fitting the data is to use the gold standard FSC. We have re-calculated the resolution of our maps using the gold standard FSC in Relion, and this gives a resolution of ~3.5 nm. We have generated a new figure showing the masked region used for gold-standard FSC calculation, and the resolution plots for the maps that we show in the main figures (Figure 1—figure supplement 2).

2) You will need to explain more clearly why certain aspects of these structures were not analysed. Mammalian centrioles are generally thought to have a distal region composed of DMTs: was this absent from the centrioles you analysed here, or was it ignored for technical reasons? Similarly, an analysis of cartwheel structure would be extremely useful to the field, yet you make no attempt to do this, even in fly centrioles where most centrioles have this structure (and it is clearly evident in the centriole shown in Figure 5A).

Doublet microtubules of the distal centriole

As pointed out by the reviewers, the transition between the TMT mammalian centriole and the DMT axoneme is an especially important region. Work from the laboratory of Michel Bornens (Paintrand et al., 1992) using serial sectioning of resin-embedded centrosomes showed how the mammalian centriole changed along it’s proximal-distal axis. Their averaging of the extreme distal region indicated a DMT, but with a curved extension (Paintrand et al., Figures 3 and 8).

The centrioles in our dataset varied in length; the average length was 440 nm, the shortest and longest being 400 nm and 480 nm, respectively. Due to this variability, we decided to use only the data common to all, which is the proximal 0 – 400 nm. In the original manuscript, we averaged along the centriole axis, grouping subvolumes into four bins of 100 nm (Figure 2).

To address the reviewer’s comments, we now have regrouped the subvolumes into smaller bins of 50 nm to get better discrimination along the centriole axis. This also allowed us to generate a new average consisting of all subvolumes in the 400 nm + region (Figure 2—figure supplement 2). This new grouping emphasizes how sharp the transition is from the pinhead to non-pinhead structure; the pinhead is a strong density up to 200 nm, and essentially absent thereafter. This fits nicely with recent data on cartwheel-length determination in the fly embryo (Aydogan et al., 2018).

The new 400 nm + average is essentially a DMT, but there is still some density for a partial C-tubule. Based on the small number of unique subvolumes in this class (N = 272), and the increase in noise in the average, we do not feel confident stating that the distal centriole is a DMT, as some partial C-tubule density persists within the extreme distal centriole average.

To better address the question on the nature of the centriole-axoneme transition in a more quantitative way, one would need to collect a larger dataset and average only centrioles of the same length, or collect data of basal bodies that have extended an axoneme. Both of these are experiments would be informative, but are beyond the scope of this current study.

The cartwheel

As noted by the reviewers, a structural analysis of the cartwheel would be extremely useful to the field. On this point we are in total agreement with the reviewers, and, as was correctly pointed out, we did not explain our reasoning for excluding it from our analysis.

We had initially wanted to analyze all components of centriole, including the cartwheel, but the flattening problem prevented that analysis. All CHO centrioles were significantly flattened, even the daughter centriole in Figure 1A.

In 2 out of 30 *Drosophila* S2 centrioles, flattening was undetectable, and it is one of those centrioles that we have used in Figures 5–7. We did try to average the cartwheels from the two non-flattened centrioles, but we were not able to get an informative average, likely due to a combination of factors. The small number of subvolumes available from two centrioles that were each ~150 nm in length was undoubtedly a limiting factor. Furthermore, the orientation of these centrioles parallel to the beam axis, while visually striking, dramatically degrades Z-resolution and results in tomograms that are sub-optimal for averaging structures that are along the Z-axis. In the 28 out of 30 centrioles that are more-optimally arranged for sub-volume averaging, all are flattened, all cartwheels are displaced, and we cannot discern them in the raw data.

3) Perhaps these omissions are due to the "flattening" problem but, if so, this needs to be better explained. Indeed, we felt it might be better to more prominently discuss the "flattening" problem at the start of the Results section, and to more directly describe how this influenced your subsequent analysis.

Centriole flattening was a major issue for our data analysis, and we accept that we did not satisfactorily address it. We have now changed the Results section by introducing the issue much earlier, and explaining why the flattening precluded an analysis of the cartwheel structure.

It is unclear to us how to overcome the centriole flattening that we observed in both the *Drosophila* S2 and CHO systems, and it dramatically affects the overall geometry of the centriole. Centrioles that are arranged parallel to the beam axis (proximal-distal axis orthogonal to the ice sheet) are not flattened and visually informative, but are missing information that contributes to the alignment and subsequent average. Based on the tomograms, it was apparent that the ice was still substantially thicker that the length of the centrioles, thus it appears to be a surprisingly long-range effect. We did collect several tomographic tilt series of centrosomes in substantially thicker ice, but we had difficulty aligning the tilt-series at tilts above 30˚. Future studies on centrioles may be better performed using ion milling, where thick samples are thinned using a focused-ion beam, preserving the geometry in a thinner sample or perhaps thick ice in conjunction with a Volta phase plate.

4) It would have greatly increased the impact of this work if you had examined the structure of fly centrioles lacking pericentrin/PLP. Your previous work showed that PLP is a key component of the interphase centriole in flies, and very likely a component of the amorphous brush-like structures you highlight in Figure 5A (Mennella et al., 2012 – see also Roque et al., 2018, which should be mentioned). We would have thought this analysis would be relatively straightforward. We would not insist on this but, if you really can't get this data, it would be helpful to discuss any insights you have as to why this approach may not be as straightforward as one might assume.

As noted by the reviewers, PLP is likely a component of the brush-like structures that we observe in the tomograms of S2 centrioles, and this fits with both light-microscopy data from *Drosophila* S2 cells (Mennella et al., 2012) and mutational analysis in *Drosophila* embryos (Roque et al., 2018).

Previous experiments showed that Plp RNAi leads to the disappearance of γ-tubulin from interphase centrosomes (Mennella et al., 2012). Unlike our mammalian centrosome preps, the *Drosophila* S2 centrosome preps tend to be contaminated with many particles that are a similar size to centrosomes, and more numerous than centrosomes. We overcame this problem by regrowing centrosome microtubules, enabling us to follow microtubules and to distinguish centrosomes from contaminants, which would be difficult with successful Plp RNAi. While this is clearly a very interesting experiment, these practical issues make it well beyond the scope of a revision.

An alternative strategy of genetically tagging proteins followed by sub-volume averaging has proven informative in defining the localization of axonemal proteins (Nicastro, Kikkawa labs and others). We have recently started a collaboration to use CRISPR to tag endogenous proteins with tags like GFP, with the aim of identifying the location of the tagged protein in the average map after sub-volume averaging. The candidates that we plan to pursue initially are those that are part of the centriole itself, and thus likely to average well and be visible in the centriole averages.

Reviewer comments informing summary review:Reviewer #2:1) The basis of averaging strategy, such as nine-fold symmetry and 24nm (3 tubulin dimers) periodicity, which is the fundamental basis of this study, is not clearly indicated. In the case of the axoneme, there are a number of studies (such as Goodenough and Heuser or by Peter Satir in 1980s) which proved 96nm repeat. In the axoneme, some structures such as MIPs follow, not 24nm, but other (16nm, 32nm) periodicity. 9fold symmetry and symmetry breaking (Hoops & Witman; Bui et al., 2012, JCB) have been described in the axoneme field. Are there any evidence that these centrioles follow 24nm periodic and 9fold symmetry? Alternative periodicity cannot be excluded to explain some weak densities. In the tomogram of vertically oriented Drosophila centrioles, it will be possible to discuss 9fold symmetry. Did they mix daughter and mother centrioles?

We have hopefully satisfactorily addressed most of these points in the ‘Major points’ section above. In addition, we have no evidence from any of our centrioles (including the end-on S2 centriole) for any symmetry other than quasi-9-fold symmetry. We do not argue that the symmetry is definitely 9-fold, but in the absence of any evidence to the contrary we assume that it is 9-fold symmetry for the purposes of regenerating the centriole geometry. Note that no strict symmetry was assumed in any of our averaging – we simply align 24 nm-long segments and average, whatever intrinsic periodicity exists is “discovered” via the unbiased alignment process.

2) Plunge-frozen centrioles used in this study were (except the one oriented vertical in ice) flattened. The authors fit averaged subtomograms from these tomograms but fitted it to another 3D tomogram which retained 9fold in the past work. Based on this fitting they discussed the positions of the A-C and A-A linkers. If these linkers keep linking the same protofilaments, they should be oriented differently in the flattened centrioles and thus should be smeared out by subtomogram averaging. What do the authors think? Do the linkers actually detach from the adjacent triplet (or doublet), retaining the angles of protrusion?

We agree that the linkers in the flattened centrioles are not adopting their correct conformation. However, the fact that all centrioles studied here are still part of an ensemble structure indicates that a majority of linkers must still be linking adjacent MTs. At high contour the MT linkers show lower occupancy in the maps than the protofilaments of the microtubules, suggestive of structural flexibility in the MT linkers. It would appear that the linkers are most coherent near where they attach to the A- and C-tubules and then become more disordered farther away.

3) Classification process is not clear. In the text, they stated that four separate structures were obtained, but only two are shown in Figure 2—figure supplement 1.Figure 2—figure supplement 1C is interesting. Do all the mammalian centrioles, which this reviewer thinks to have variety of the entire length, share the same length of the proximal region? How do they explain significant amount of "red" subtomograms within the white area in Figure 2—figure supplement 1C? Do they appear in all the centrioles analyzed in this work?

We have added more to the Materials and methods section about classification and clarified the main text: “As a first step, unbiased 3D classification (Bharat and Scheres, 2016) suggested two major structural-distinct classes arranged differentially along the centriole length (Figure 2—figure supplement 1). While the classification was somewhat noisy, likely due to a combination of biological and imaging noise, it indicated changes along the proximal-distal axis, in agreement with previously published data (Paintrand et al., 1992)”.

Major point 2 deals with centriole length, and the invariant length of the proximal region.

Reviewer #3:1) Doublet microtubule region of the centriole: The authors extracted the triplet microtubule region of centrioles for subvolume averaging (subsection “Mammalian centriole structure”). However, it is well-known that mature (as these seem to be) mammalian centrioles have a distal region composed of doublet microtubules (Paintrand et al., 1992). Analysis of this region is especially important to understanding the transition between the centriole and ciliary axoneme. The authors do describe differences that appear between the proximal end of the centriole and the region where the C-tubule becomes only a partial tubule, but it would helpful to know whether they averaged the doublet microtubule region proper, to determine whether there may be similarities or differences between the doublet and triplet microtubule regions. If not, this should be clearly stated in the text, in addition to a rationale about why this region was not analyzed. This is potentially very important in understanding whether the mature mammalian centriole could more properly be thought of as a core centriole with an axoneme initial segment on its distal end.

We agree with the issues raised here and hope we have addressed them in the major points section, and with the accompanying Figure 2—figure supplement 2.

2) The cartwheel: The authors state that due to the centriole cycle, it is likely that none of the CHO cell centrioles retained their cartwheels. However, only centrioles in G1 lack cartwheels – procentrioles in S-phase, G2, and even early mitosis have cartwheels. In addition, centriole pairs in which the mother and procentriole are orthogonal and close to each other are probably engaged and thus be at a stage prior to removal of the cartwheel after disengagement (Shukla et al., 2015, Nature Communications). Thus, the centriole pair in Figure 1A is likely engaged based on orientation and proximity of the two centrioles. Did the authors analyze the base of this daughter centriole (or procentriole, if still engaged) to determine whether there is a cartwheel present? Were there other orthogonal pairs in their dataset that may have cartwheels in the procentrioles? This would be especially important to understand the nature of centriole engagement, for which no structural correlates are known. If no cartwheel was found in the orthogonal centriole pair(s), the authors should note that there is a possibility that the cartwheels were lost from CHO cell centrioles during isolation and fixation. Drosophila centrioles, in contrast to mammalian centrioles, retain their cartwheel though the cell cycle, so one would expect that all fly centrioles examined have, or had, cartwheels, Consistent with this, the fly centriole shown in Figure 5A has a cartwheel, based on the presence of a central tubule, which is likely the hub of the cartwheel. Was the cartwheel retained in all fly centrioles examined? If not, it would be important to note that the preparation results in loss of some cartwheel structure. This might also allow a determination as to whether it is retention of the cartwheel that better allowed some Drosophila centrioles to retain their structure and overall geometry.

Again, we are in agreement with this and hope we have satisfactorily addressed the concerns. While we wanted to examine the cartwheel in centrioles and compare it with that from basal bodies, the data did not allow us to do so.

3) Evolution: The authors previously published structural analysis of Chlamydomonas centrioles, which, like mammalian centrioles, have triplet microtubules. To maximize the evolution impact of this work, it would be helpful to know whether the linkages that reinforce 9-fold symmetry are similar between Chlamydomonas and mammalian centrioles, as well as whether any of the non-microtubule densities observed in C.r. are similar to those described here. In addition, the authors note that the density spanning A09 to A10 is conserved between all centriole and axoneme structures – could they state more definitively the degree to which the shape of this structure is similar between species, and thus whether it is consistent with a similar protein (complex) forming this structure?

While a MIP at the A09/A10 position has also been identified in axonemes (MIP2), we cannot draw direct structural comparisons between it and the A09/A10 MIP in CHO and S2 centrioles. At our resolution our MIP appears to be a dimer of two similarly-sized proteins that bind to each other and to the microtubule lattice across the microtubule seam. High-resolution work from Ichikawa et al. (2017) shows MIP2 binding an analogous location, but our resolution is much too low to allow a more-detailed comparison.

4) Partial B-tubules: The authors interpret the presence of partial B-tubules that lack protofilaments B03-B06 as evidence that B-tubule assembly in Drosophila occurs from both the inner and outer edges of the A-tubule. However, it is known that during thermal fractionation of axonemes, B-tubule stubs attached to the A-tubule remain, while the rest of the B-tubule is lost (Linck and Langevin, 1981, JCB; Linck et al., 2014). Can the authors rule out the possibility that the partial B-tubule is a degradation product? If not, this interpretation should either be removed or qualified accordingly.

No, we cannot rule out depolymerisation/damage as a cause for the microtubule stubs in the S2 centriole, and so we have included the caveats as suggested by the reviewer (subsection “Biogenesis of the B-tubule”).